# Immunomodulatory Effects of Traditional Korean Gochujang in Rats Immunosuppressed with Cyclophosphamide

**DOI:** 10.3390/ijms26178325

**Published:** 2025-08-27

**Authors:** Hak Yong Lee, Young Mi Park, Dong Yeop Shin, Hai Min Hwang, Sung Hak Chun, Sang Jin Lim, Hee-Jong Yang, Gwang Su Ha, Myeong Seon Ryu, Ji-Won Seo, Do-Youn Jeong, Jun Sang Bae, Jae Gon Kim

**Affiliations:** 1INVIVO Co., Ltd., 121, Deahak-ro, Nonsan 32992, Republic of Korea; leeapf@nate.com (H.Y.L.); pym07130@hanmail.net (Y.M.P.); 9909867@hanmail.net (S.H.C.);; 2Department of Pathology, College of Korean Medicine, Wonkwang University, Iksan 54538, Republic of Korea; 3Microbial Institute for Fermentation Industry (MIFI), Sunchang-gun 56048, Republic of Korea

**Keywords:** *gochujang*, immune-enhancing, spleen, cytokines, NK cell, MAPKs/NFκB pathway

## Abstract

Fermented foods are consumed in several cultures worldwide and their health benefits are being increasingly reported. Fermented soybean products in Asia include soybean paste (*doenjang*), fermented soybeans (*cheonggukjang*), red pepper paste (*gochujang*, GCJ), and natto. These fermented foods are reportedly associated with health benefits, including the alleviation of colitis and improvement of immune function. In this study, we investigated the immune-enhancing effects of GCJs produced in four different regions of Korea (including a commercial brand) in cyclophosphamide-treated immunosuppressed rats. The GCJs-treated group showed prevention of weight loss, increased levels of butyric acid in the cecum, and increased weight of lymphoid organs such as the thymus and spleen. Whole blood and serum analysis revealed increased numbers of white blood cells, including granulocytes and lymphocytes, pro-inflammatory cytokines (IL-2, IL-12, IFN-γ, and TNF-α), and elevated immunoglobulin G levels. Additionally, splenocyte proliferation, splenic natural killer cell activity, and immune-related signal pathways (MAPK and NF-κB) were increased. Histological analysis revealed improved tissue structure in the GCJs group. In conclusion, these findings show that GCJs enhance immune function by promoting the growth of immune organs, increasing cytokine production, and activating immune-related signaling pathways. These results suggested the potential of traditional Korean GCJs as a dietary intervention to improve immune health.

## 1. Introduction

Immunity, an important component of cell survival, is responsible for defending the body against harmful viruses and substances that penetrate it [1]. Reduced immunity can occur due to chemical exposure (e.g., pharmacological treatments for rheumatoid arthritis, atopy, and cancer immunotherapy), aging, malnutrition, and extreme stress [2]. The immune system comprises innate and adaptive immunity, which depend on immune organs, such as the thymus and spleen, and immune cells (including macrophages, natural killer (NK) cells, and splenocytes), which play important roles in enhancing the body’s immunity [1,3]. NK cells and splenocytes are indicators of immunity. NK cells exert direct cytotoxic effects on cancer cells and stimulate cytokine production. Increased splenocyte proliferation can upregulate immunity [4]. In a previous study, a 4-week intake of *Cheonggukjang* (CGJ) increased immunity by enhancing cytokine secretion, NK cell activity, and splenocyte proliferation in cyclophosphamide (CP)-treated immunosuppressed rats. Accordingly, cytokine production, NK cell activity, and splenocyte proliferation were investigated to assess immune enhancement [3].

Globally, fermented foods and sauces have long existed in various forms (solid and liquid) [5]. Cheese and wine are consumed in the West, whereas *gochujang* (fermented red pepper paste; GCJ), CGJ, *doenjang* (fermented soybean paste), and natto (in Japan) are consumed in the East [6]. Previous studies have reported that fermented soybean products are rich in bioactive compounds such as peptides and isoflavones, which may contribute to various therapeutic effects. In particular, during the fermentation process of GCJ, key functional components such as capsaicin, daidzin, daidzein, and genistein were not significantly altered in their content. However, extracts derived from GCJ have been shown to exhibit stronger inhibitory activity against β-glucosidase, suggesting a potential enhancement of bioactivity associated with fermentation [7,8]. It also contains several beneficial components. Capsaicin, the active component of chili pepper, possesses anti-inflammatory properties and modulates the immune response by influencing cytokine production and enhancing immune cell activity [9,10]. Daidzein and daidzein, which are isoflavones found in soy products, can enhance immune function through various mechanisms, including the modulation of the gut microbiota [11,12]. Genistein, another soy isoflavone, lowers blood glucose and lipid levels and exerts anti-inflammatory and cancer chemopreventive effects [11,13]. *Lactobacillus* can improve the overall immune function and reduce inflammation by modulating the gut microbiota [14,15]. Therefore, fermented foods are effective in preventing cardiovascular and metabolic diseases and can improve intestinal immunity, as well as brain and skin health.

GCJ is a traditional Korean fermented sauce made from gochu powder (dried chili peppers), meju powder (fermented soybean powder), and salt [16]. It contains bioactive compounds such as capsaicin, genistein, and daidzein, and various health benefits of GCJ have been reported [8]. The anti-inflammatory properties of dextran sulfate sodium (DSS)-induced colitis have been demonstrated in inflammatory disease models [16,17]. However, the immune-enhancing effects of GCJ have not been fully investigated. Therefore, in the present study, we examined the effects of 4-week oral administration of GCJ sourced from various regions of Korea on immune function in Wistar rats with cyclophosphamide (CP)-induced immunosuppression.

## 2. Results

### 2.1. Component of GCJs

The pungent compounds in GCJs are capsaicinoids, primarily capsaicin and dihydrocapsaicin. In this study, the total capsaicin content was determined as the sum of capsaicin and dihydrocapsaicin. Four samples of GCJs (S1–S4) were analyzed using HPLC to quantify the total capsaicin content. The results are presented in Table 1.

### 2.2. Animal Monitoring and SCFAs Analysis

BW was monitored weekly following the administration of CP (5 mg/kg), GCJs (500 mg/kg, S1–S4), or the positive control HemoHIM (1000 mg/kg). The vehicle group, which received CP only, exhibited a gradual decrease in BW throughout the experimental period compared with the other groups. At the end of the experiment, all GCJs (S1–S4)-treated groups showed a significantly higher BW than the vehicle group (Figure 1A). After the autopsy, the weights of the immune-related organs (thymus and spleen) were measured (Figure 1B,C). Thymus weight was significantly reduced in all CP-treated groups compared with that in the normal group. However, GCJs (S2–S4)-treated and positive control groups exhibited an increase or significant recovery in thymus weight compared to that in the vehicle group (Figure 1B). Similarly, the spleen weight was reduced in the CP-treated groups relative to that in the normal group. In contrast, significant increases were observed in all GCJs (S1–S4)-treated groups and the positive control group (Figure 1C). We analyzed short-chain fatty acids (SCFAs), including acetic acid, propionic acid, and butyric acid, in cecal contents collected immediately after necropsy to further evaluate the intestinal immune response (Figure 1D,F). Acetic acid and propionic acid levels were not significantly different between the groups (Figure 1D,F). However, butyric acid, which was reduced in the vehicle group following CP treatment, was significantly increased in GCJs (S1–S4)-treated groups (Figure 1F). Notably, the butyric acid levels were not significantly increased in the positive control group. These findings suggested that GCJ intake under immunosuppressive conditions alleviates immune dysfunction by preventing weight loss and increasing butyric acid levels in the gut.

### 2.3. Blood and Serum Analyses

The immune system is mediated by immune cells, cytokines, and immunoglobulins [18]. Immune cells (white blood cells (WBC), granulocytes, and lymphocytes), cytokines (interleukins, interferon (IFN)-γ, and tumor necrosis factor (TNF)-α), and immunoglobulin G (IgG), which are responsible for immunity, were reduced in immunosuppressed animals [19]. Whole blood was collected from the vena cava; one portion was used to perform CBC measurements, and the other portion was separated to obtain serum (Figure 2). CBC was performed immediately after collecting whole blood, and the vehicle group (only CP-treated) showed significantly reduced levels of WBC, granulocytes, lymphocytes, and mid-sized cells compared to those in the normal group (Figure 2A–D). In contrast, GCJs (S1–S4)-treated groups showed an increase in WBC, granulocytes, and lymphocytes, although the levels of mid-size cells remained unchanged. In particular, the GCJ S2, GCJ S4, and positive control groups exhibited significant increases in CBC levels compared to the vehicle group (Figure 2).

Levels of cytokines (including interleukin (IL)-2, IL-12, IFN-γ, and TNF-α) and IgG were analyzed in the separated serum (Figure 3). The vehicle (CP only) group demonstrated a significant reduction in cytokine and IgG levels compared to the normal group, whereas the levels were enhanced in the GCJs (S1–S4)-treated groups. IL-2 levels were elevated in the S1- and S4-treated groups, and a significant increase was observed in the S2- and S4-treated groups (Figure 3A). IL-12 and IFN-γ levels were significantly increased in the GCJs (S1–S4) groups (Figure 3B,C). The TNF-α level was significantly elevated in the GCJs (S2–S4)-treated groups, except for the S1-treated group (Figure 3D). Furthermore, IgG levels significantly increased in the S2-treated group (Figure 3E). These results suggested that GCJ consumption could facilitate the maintenance of immune function by regulating the components involved in immunity in both whole blood and serum.

### 2.4. Splenocyte Proliferation and NK Cell Activity

The spleen plays a crucial role in regulating immune responses by serving as a site where immune cells such as T cells, B cells, and macrophages are localized in response to antigens [20]. Splenocyte proliferation is an important indicator of immune response and can help measure the strength of immune function. Cells were treated with lipopolysaccharide (LPS) and concanavalin A (ConA) to examine splenocyte proliferation (Figure 4). Following treatment with LPS and ConA for 24 h, splenocyte proliferation was significantly reduced in the CP-treated group compared with that in the normal group.

In the GCJs (S1–S4)-treated group, treatment with LPS for 24 h increased splenocyte proliferation in the S1, S2, and S3-treated groups compared with that in the control group, although the difference was not significant. Only the S4-treated group exhibited a significant increase in splenocyte proliferation (Figure 4A). Following ConA treatment for 24 h, splenocyte proliferation was significantly increased in the GCJs (S1, S2, and S4)-treated groups compared to that in the CP-treated group. Although the S3-treated group showed an increase, it was not statistically significant (Figure 4B).

The spleen plays a notable role in regulating NK cell activity as it is a major site for NK cell activation and functions in immune responses. NK cell activity was measured in cultured splenocytes by assessing the cytotoxicity against target cells (AR42J cell line). Compared with the CP-treated group, the GCJs (S1–S4)-treated groups showed significantly increased NK cell activity (Figure 4C).

### 2.5. MAPKs/NFκB Pathway

Using harvested splenic tissues, we examined the phosphorylation of mitogen-activated protein kinases (MAPKs; Erk, JNK, and p38) and nuclear factor-kappa B (NFκB) in regulating immune mechanisms (Figure 5). The phosphorylation of MAPKs and NFκB was significantly decreased in the CP-treated group compared with that in the normal group. Conversely, phosphorylation of MAPKs and NFκB was increased in the GCJs (S1–S4)-treated groups compared with that in the CP-treated group (Figure 5B–D). These results suggested that CP suppresses immune mechanisms and that the intake of GCJs could induce the activation of these mechanisms, thereby enhancing immune function.

### 2.6. Histological Analysis

The spleen is divided into white pulp (WP), where T and B cells are activated to produce antibodies or induce immune responses, and red pulp (RP), which is responsible for the removal of damaged red blood cells [21]. In the WP, which plays a crucial role in immune response, antibodies and immune cells are transformed into memory T cells. These memory T cells enable a rapid and strong immune response upon invasion by the same pathogen [22]. The immunosuppressant (control group), CP, induced atrophy of WP in the spleen and sporadic cell condensation (Figure 6), leading to the collapse of the marginal zone. Consequently, the distinction between WP and RP is unclear, and lymphoid depletion was observed owing to the irregular cellular arrangement of RP. The GCJs (S1–S4)-treated and positive control groups showed improvement in WP compared with the CP-treated group. The S1- and S4-treated groups showed localized cell condensation within the WP, although this was improved compared to the CP-treated group. Furthermore, in the S2- and S3-treated and positive control groups, RP condensation and WP atrophy were reduced, and the marginal zone showed improvement.

## 3. Discussion

Decreased immunity can be attributed to various factors, and a persistent reduction in immunity can lead to a combination of symptoms, including increased susceptibility to bacterial and viral infections, weight loss, loss of appetite, fatigue, and fever [23]. Thus, preserving and strengthening the immune system play vital roles in maintaining good health. CP, a widely used immunosuppressive agent, has been used in previous studies to induce immunosuppression. Oral administration or intraperitoneal injection of CP has been shown to reduce BW and the weights of immune-related organs such as the thymus and spleen. In addition, CP decreases the number of circulating immune cells, including WBC, granulocytes, and lymphocytes. It has also been reported to suppress the immune signal pathway (MAPKs and NFκB) in the spleen, inhibit splenocyte proliferation and splenic NK cell activity, and disrupt normal splenic tissue structure [3,19]. Therefore, CP-induced immunosuppressive models have been widely used to evaluate the efficacy of immune-enhancing agents and extracts.

Our study investigated the immune-enhancing effects of traditional Korean GCJs by assessing their impact on immune biomarkers (BW, CBC, serum analysis, splenic NK activity, splenic proliferation, and immune signaling pathways) in a CP-induced immunosuppressed model. In addition, SCFAs (such as acetic acid, propionic acid, and butyric acid), metabolites of gut microbiota, play crucial roles in immune function by influencing immune cell migration, adhesion, and cellular functions (such as proliferation and apoptosis), thereby regulating immune responses [24,25]. Therefore, given that decreased immunity and intestinal butyrate levels are closely associated, changes in SCFA content were examined. In our results, the groups treated with GCJs (S1–S4) showed a significant increase in BW compared to that in the CP-treated group (vehicle group). As a decrease in BW is one of the earliest observable signs of immune suppression, this finding suggested that food intake may help enhance immune function by alleviating the weight loss associated with immune decline. Complete blood count (CBC) analysis revealed elevated WBC, granulocyte, and lymphocyte counts. An increase in BW and CBC indicates an active immune response, suggesting that the immune function is enhanced under immunosuppressed conditions. Under conditions of decreased immunity, an increase in butyrate levels due to GCJs consumption can improve the immune response. Butyric acid promotes immune cell activation and antibody production, thereby contributing to the restoration of immune function [26]. These results suggested that increased butyrate levels in an immunocompromised state can positively affect immune enhancement.

Serum cytokines play important roles in the immune system. Cytokines increase T cell and NK cell activity, promote T helper 1 response, and enhance immune function by inducing apoptosis in infected and tumor cells [27]. In this study, we determined the levels of IL-2, IL-12, IFN-γ, and TNF-α in the serum of animals fed GCJ to assess the immune-enhancing effect of GCJs. Serum levels of IL-2, IL-12, IFN-γ, and TNF-α were increased in the GCJs (S1–S4)-treated group. Elevated serum concentrations of these cytokines correlated with a stronger immune response, highlighting the immune-enhancing effects of GCJs.

The spleen is closely related to immune enhancement and plays an important role in various immune functions, such as immune cell activation, antibody production, pathogen elimination, and the regulation of immune responses [19,20,21]. It also contributes considerably to strengthening the defense against infections and the formation of immune memory. The spleen plays a key role in regulating immune cell activation and antibody production and serves as an important organ in which lymphocytes, especially B and T cells, are activated and interact [28]. The proliferation of splenocytes indicates that the immune system recognizes and responds to invading pathogens and can induce a faster and more robust immune response upon re-exposure to the same pathogen [29]. Therefore, the increased proliferation capacity of splenocytes is an important indicator of the immune system status. In this study, splenocyte proliferation was evaluated after stimulation with LPS and ConA to induce humoral and cellular immune responses, respectively, and assess the efficacy of GCJs for immune enhancement. The results revealed that while the proliferative capacity of splenocytes decreased in the CP-treated group, splenocytes from animals that consumed GCJs exhibited increased proliferation compared with those from CP-treated rats. Accordingly, the increased splenocyte proliferation in the GCJs (S1–S4)-treated group compared with that in the CP-treated group indicates that the traditional Korean GCJs have immune-enhancing effects.

The MAPK and NF-κB signaling pathways play important roles in immune enhancement. These pathways are involved in the activation, proliferation, differentiation, and inflammatory responses of the immune cells [30,31,32]. The activation of these pathways regulates immune cell proliferation, activation, and inflammatory responses, thereby strengthening defense against pathogens. In this study, phosphorylation of MAPKs and NF-κB in the spleen of CP-treated animals was reduced compared with normal levels, whereas restored activation was observed in the spleen tissues of GCJs-fed animals. These results suggested that the intake of GCJs (S1–S4) can activate the MAPK and NF-κB pathways, thereby enhancing the function of immune organs and potentially playing a critical role in immune regulation. Finally, histological examination of the spleen tissue revealed that the spleen structure of the GCJs (S1–S4)-treated group was significantly improved compared with that of the CP-treated group.

In conclusion, our findings revealed that traditional Korean GCJs (S1–S4) exerted immune-enhancing effects in CP-induced immunosuppressed rats. The intake of GCJs enhanced gut butyrate, BW, WBC, and cytokine levels. Furthermore, splenocyte proliferation was increased in the GCJs (S1–S4)-treated group compared with that in the CP-treated group, and the activation of the MAPK and NF-κB pathways was restored. Histological improvement in the spleen tissue was also confirmed. Collectively, these findings suggest that the traditional Korean GCJs (S1–S4) positively influence immune enhancement under immunocompromised conditions.

## 4. Materials and Methods

### 4.1. Preparation of GCJs

Korean GCJs were provided by the Microbial Institute for the Fermentation Industry (MIFI, Sunchang-gun, Republic of Korea). The four types of GCJs were as follows: S1, Jeju-si (Jeju Special Self-Governing Province, Republic of Korea); S2, Sunchang-gun (Jeollabuk-do Province, Republic of Korea); S3, Yangyang-gun (Gangwon Special Self-Governing Province, Republic of Korea); and S4, commercial brand GCJ (Jeollabuk-do Province, Republic of Korea). GCJs (S1–S4) were prepared according to the recipe for each region of Korea. GCJs were finely ground in distilled water (DW; 150 and 350 g) using a blender for 60 s. To determine the dry weight, 3 mL of each diluted GCJs were dried in a drying oven at 60 °C for 24 h. After drying, the weight of the dried product was measured (S1: 170.0 ± 8.1 mg/mL, S2: 185.4 ± 0.8 mg/mL, S3: 216.4 ± 1.8 mg/mL, and S4: 188.8 ± 0.8 mg/mL); GCJs were orally administered daily at 500 mg/kg based on the dry weight.

### 4.2. High Performance Liquid Chromatography of GCJs

Standard solutions of capsaicin and dihydrocapsaicin were prepared by dissolving 5 mg of each compound in methanol and diluting to 50 mL. Standards were prepared at concentrations of 0.1, 0.5, and 1 mg/kg by appropriate dilution of the standard solutions. These were used to generate calibration curves for quantification. A 0.2 g of GCJs (S1–S4) was extracted with 15 mL of 100% methanol at 90 °C for 1 h with intermittent shaking, cooled to room temperature, adjusted to 25 mL with methanol, and filtered through a 0.2 μm membrane prior to analysis. Component analysis of standard solutions and GCJs (S1–S4) was performed using a Semi-Micro HPLC system (Shiseido, Tokyo, Japan) equipped with a fluorescence detector. Separation was carried out on an Agilent Eclipse XDB-C18 column (3.0 × 75 mm, 3.5 μm). The mobile phase consisted of 1% acetic acid in water and acetonitrile in a ratio of 60:40 (*v*/*v*), without gradient elution. The flow rate was set at 1.0 mL/min, and the column temperature was maintained at 40 °C. The injection volume was 10 μL. The final solution was filtered through a 0.2 μm membrane filter prior to HPLC analysis.

### 4.3. Animals and Oral Administration

All the experiments were conducted in accordance with the National Institutes of Health Guidelines for the Care and Use of Animals. This study was approved by the Institutional Animal Care and Use Committee of INVIVO Co., Ltd. (Nonsan, Republic of Korea; IV-RB-17-2405-05 and date of approval of 30 April 2024). Male Wistar rats (5 weeks old) were purchased from Orient BIO Co. (Seongnam-si, Gyeonggi-do, Republic of Korea) and divided into seven groups (normal, control, S1, S2, S3, S4, and positive control; Table 2). The rats were orally administered CP (5 mg/kg) to induce immunosuppression. Concurrently, the rats received either GCJs (500 mg/kg) or HemoHIM herbal preparation (1000 mg/kg, positive group; purchased from Kolmar BNH Co., Ltd., Sejong, Republic of Korea) via oral gavage for 4 weeks. HemoHIM was selected as a positive control, as previous studies have demonstrated its immune-enhancing effects [19]. The body weight (BW) of the animals was monitored weekly during the experimental period. During autopsy, the weights of the thymus and spleen were measured immediately.

### 4.4. Short-Chain Fatty Acid Analysis

Quantitative analysis of short-chain fatty acids (SCFAs), specifically acetic, propionic, and butyric acids, was performed. Cecal stool (0.1 g) from the rats was mixed with 0.25 mL of 1 N HCl and 0.025 mL of 0.1 M isobutanol (used as an internal standard), followed by agitation for 10 min. Fatty acids were extracted using diethyl ether. The mixture was centrifuged at 12,000 rpm for 5 min, and the supernatant was collected for SCFA analysis. A 1 μL aliquot of the supernatant was injected into a gas chromatograph–mass spectrometer (GC-MS, Hewlett Packard Model 7890; Agilent Technologies, Palo Alto, CA, USA) equipped with a DB-FATWAX Ultra Inert column for quantification.

### 4.5. Complete Blood Cell (CBC) Count Analysis

Following respiratory anesthesia, whole blood was drawn from the vena cava and transferred to tubes pre-coated with ethylenediaminetetraacetic acid (EDTA). Complete blood count (CBC) was immediately analyzed using a hematology analyzer (BC-2800; Mindray, Bath, UK).

### 4.6. Serum Levels of Cytokines and Immunoglobulin G (IgG)

Whole blood was collected from the vena cava and placed in conical tubes without anticoagulants. The samples were allowed to clot at room temperature for 30 min, after which serum was separated by centrifugation at 3000 rpm for 15 min at 4 °C. Serum concentrations of IL-2, IL-12, IFN-γ, TNF-α, and IgG were quantified using commercially available ELISA kits (IL-2: MBS269718, Mybiosource; IFN-γ: ab239425, Abcam, Cambridge, UK; TNF-α: CSB-E11987r, CUSABIO; IgG: ab189578, Abcam). Absorbance was measured using a Sunrise microplate reader (Tecan, Männedorf, Switzerland).

### 4.7. Primary Cell Culture and Proliferation of Splenocytes

Splenic tissues were gently crushed using needles and then passed through a cell strainer (70 μm, SPL Life Sciences, Pocheon-si, Republic of Korea) to culture Wistar rat splenocytes. The collected splenocytes were washed with RPMI-1640 medium (Invitrogen, Carlsbad, CA, USA), and red blood cells were removed using red blood cell lysis buffer (Sigma-Aldrich, Carlsbad, MO, USA). The splenocytes were maintained in RPMI-1640 supplemented with 10% fetal bovine serum and 1% penicillin-streptomycin (10,000 U/mL; Invitrogen) in a 5% CO_2_ incubator. After 24 h of splenocyte stabilization (5 × 10^5^ cells/well), cell proliferation was measured using a WST-1 Assay Kit (ITSBio, Seoul, Republic of Korea) and an ELISA plate reader (Tecan, Männedorf, Switzerland).

### 4.8. Splenic NK Cell Activity

AR42J cells (CRL-1492), a target cell line, were purchased from the American Type Culture Collection (Innovation, VA, USA) to assess splenic NK cell activity. Primary splenocytes were isolated from all the groups (normal, control, GCJ-treated, and GCJ-positive) and used as effector cells. These cells were co-cultured in 96-well plates at a 25:1 effector-to-target cell ratio in a 5% CO_2_ incubator at 37 °C for 24 h. The viability of the AR42J cells was measured using a WST-1 Assay Kit and an ELISA plate reader. NK cell activity was calculated as the survival rate of AR42J cells compared with that of the normal group.

### 4.9. Tissue Lysis and Western Blot Analysis

For spleen tissue disruption, harvested splenic tissues were lysed with PRO-PREP Protein Extraction Solution (Cat No. 17081; iNtRON) containing complete protease inhibitor cocktail (Roche, Basel, Switzerland). Total protein (10 μg/well) was separated using 10% sodium dodecyl sulfate-polyacrylamide gel electrophoresis, transferred to PVDF membranes (Bio-Rad, Hercules, CA, USA), blotted, and incubated with the specific antibodies, including Phospho-Erk (#9101), Erk (#9102), Phospho-p38 (#9216), p38 (#9212), Phospho-JNK (#4671), JNK (#9252), Phospho-NFκB (#3033), nuclear factor kappa B (NF-κB; #8242), and β-actin (#4857), which were purchased from Cell Signaling Technology (Danvers, MA, USA). Target band intensities were detected using a Western blot imaging system (Azure Biosystems c300) and quantified using the ImageJ software 1.53k (National Institutes of Health, Bethesda, MD, USA).

### 4.10. Histologic Analysis of the Spleen

Briefly, spleens were excised from Wistar rats, weighed, and immersed in 10% neutral-buffered formalin for fixation. After 48 h of fixation, the tissues were embedded in paraffin and sectioned into 4 μm slices using a microtome (Thermo Scientific, Waltham, MA, USA). Tissue sections were stained with hematoxylin and eosin (H&E) and imaged using a Motic EasyScan One digital slide scanner (Motic, Hong Kong, China).

### 4.11. Statistical Analysis

All data are expressed as the mean ± standard error of the mean, and differences between groups were analyzed using one-way ANOVA (Duncan’s multiple-range test). All statistical analyses were performed using SPSS software (version 23.0; IBM Corp., Armonk, NY, USA). Each value represents the mean of at least three independent experiments for each group. Statistical significance was set at *p* < 0.05. Superscript letters indicate differences among groups: groups sharing the same letter are not significantly different, whereas groups with different letters are significantly different.

## Figures and Tables

**Figure 1 ijms-26-08325-f001:**
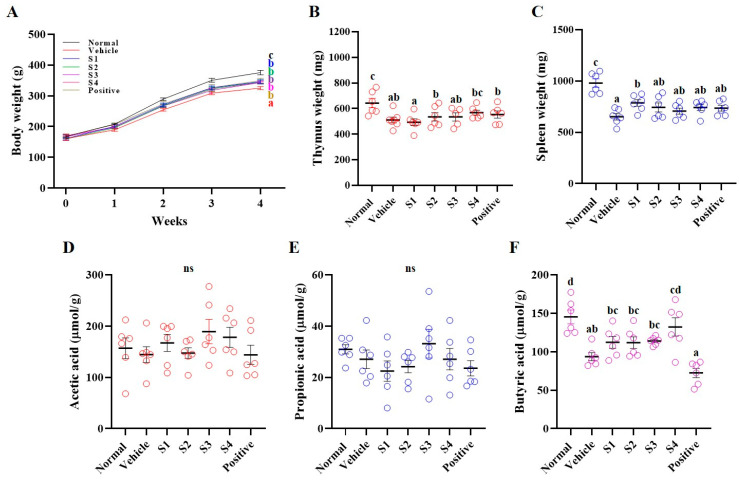
Effects of *Gochujang* on body weight and short-chain fatty acids of immunosuppressed rats. (**A**) Body weight (black line, normal group; red line, vehicle group; blue line, S1 group; green line, S2 group; purple line, S3 group, pink line, S4 group; brown line, positive group), (**B**) thymus weight, (**C**) spleen weight, (**D**) acetic acid, (**E**) propionic acid, and (**F**) butyric acid. Values in the row with different superscript letters are significantly different, *p* < 0.05; *n* = 6. Abbreviations: Normal, untreated group; Vehicle, only CP (5 mg/kg) treated group; S1–S4, CP with GCJ (500 mg/kg) treated groups; Positive, CP with HemoHim (1000 mg/kg) treated group.

**Figure 2 ijms-26-08325-f002:**
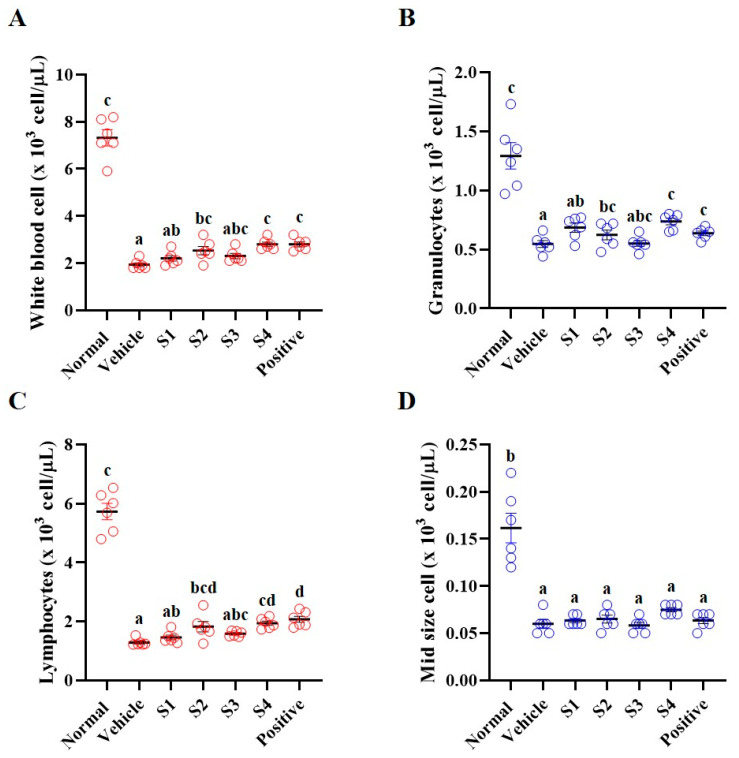
Effects of *Gochujang* on the whole-blood of immunosuppressed rats. (**A**) White blood cells, (**B**) granulocyte, (**C**) lymphocyte, and (**D**) mid-size cell Values in the row with different superscript letters are significantly different, *p* < 0.05; *n* = 6. Abbreviations: Normal, untreated group; Vehicle, only CP (5 mg/kg) treated group; S1–S4, CP with GCJ (500 mg/kg) treated groups; Positive, CP with HemoHim (1000 mg/kg) treated group.

**Figure 3 ijms-26-08325-f003:**
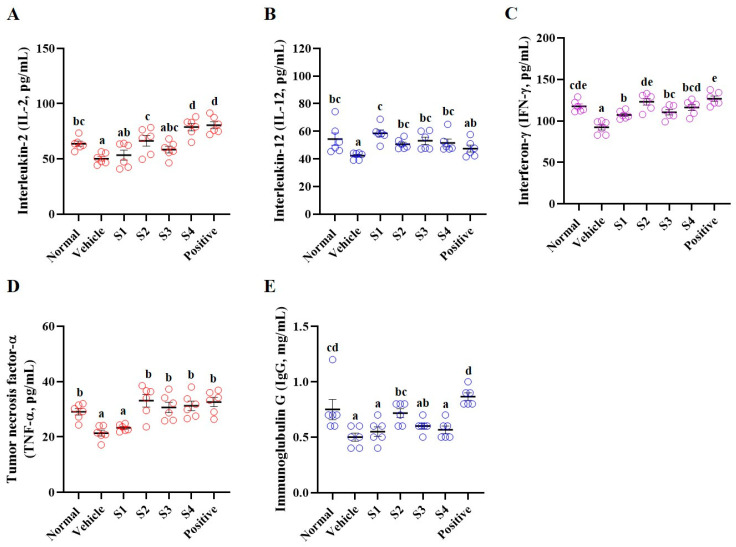
*Gochujang* increases the level of pro-cytokines in immunosuppressed rats. Level of (**A**) IL-2, (**B**) IL-12, (**C**) IFN-γ, (**D**) TNF-α, and (**E**) IgG in serum. Values in the row with different superscript letters are significantly different, *p* < 0.05; *n* = 6. Abbreviations: Normal, untreated group; Vehicle, only CP (5 mg/kg) treated group; S1–S4, CP with GCJ (500 mg/kg) treated groups; Positive, CP with HemoHim (1000 mg/kg) treated group.

**Figure 4 ijms-26-08325-f004:**
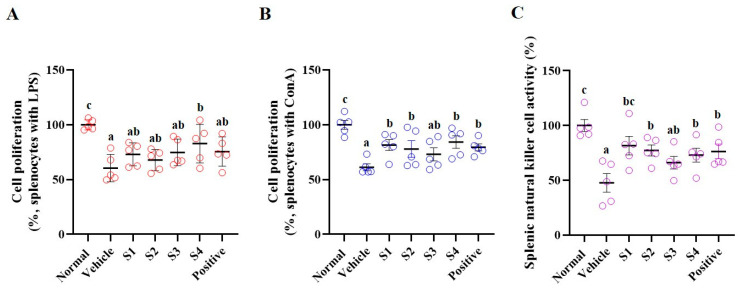
*Gochujang* activates cell proliferation and NK cell activity in the spleen of immunosuppressed rats. (**A**) cell proliferation by LPS, (**B**) cell proliferation by ConA, (**C**) NK cell activity. Values in the row with different superscript letters are significantly different, *p* < 0.05; *n* = 5. Abbreviations: Normal, untreated group; Vehicle, only CP (5 mg/kg) treated group; S1–S4, CP with GCJ (500 mg/kg) treated groups; Positive, CP with HemoHim (1000 mg/kg) treated group.

**Figure 5 ijms-26-08325-f005:**
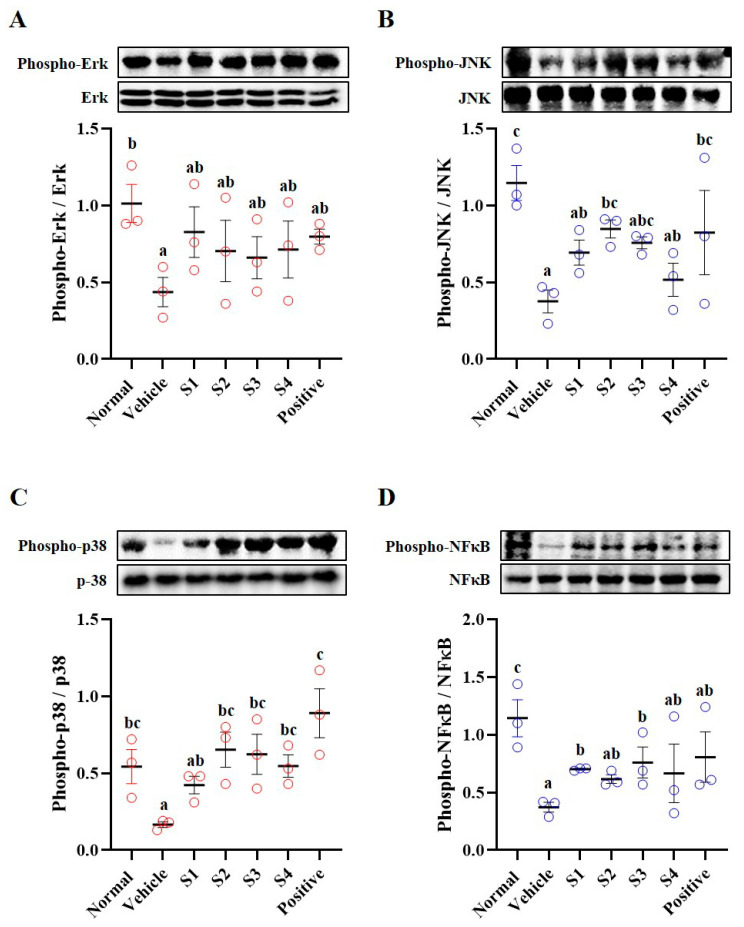
*Gochujang* increases immune-related signal pathways in the spleen of immunosuppressed rats. Phosphorylation of (**A**) Erk, (**B**) JNK, (**C**) p38, and (**D**) NFκB. Values in the row with different superscript letters are significantly different, *p* < 0.05; *n* = 3. Abbreviations: Normal, untreated group; Vehicle, only CP (5 mg/kg) treated group; S1–S4, CP with GCJ (500 mg/kg) treated groups; Positive, CP with HemoHim (1000 mg/kg) treated group.

**Figure 6 ijms-26-08325-f006:**
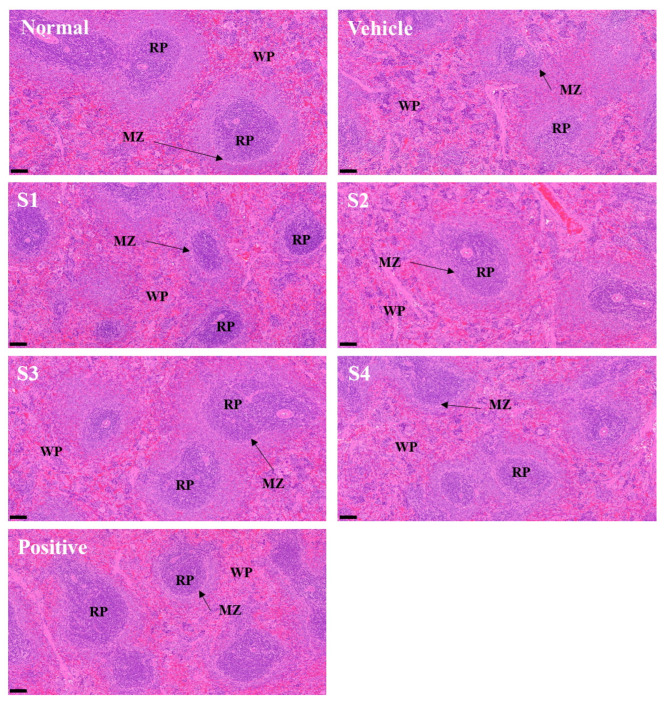
*Gochujang* improves the histological shape of the spleen in immunosuppressed rats. Abbreviations: Normal, untreated group; Vehicle, only CP (5 mg/kg) treated group; S1–S4, CP with GCJ (500 mg/kg) treated groups; Positive, CP with HemoHim (1000 mg/kg) treated group; RP, red pulp; WP, white pulp; and MZ, marginal zone (arrow); H&E stain magnification, 10×; scale bar = 100 μm.

**Table 1 ijms-26-08325-t001:** Capsaicin of Korean GCJs.

GCJs Region in Korea	S1 Jeju-si	S2 Sunchang-gun	S3 Yangyang-gun	S4 Commercial
Total capsaicin (μg/kg)	86.75 ± 1.78	139.21 ± 2.75	77.92 ± 1.38	13.67 ± 0.52

**Table 2 ijms-26-08325-t002:** Oral administration of Korean GCJs.

Groups	Oral Administration for 4 Weeks
Normal group	Untreated group
Vehicle group	Only CP (5 mg/kg)-treated group
GCJs (S1–S4) treated groups	CP (5 mg/kg) with GCJs (S1–S4; 500 mg/kg) treated groups
Positive (HemoHim) treated group	CP (5 mg/kg) with HemoHim (1000 mg/kg) treated group

## Data Availability

Data will be made available upon request.

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
