# Peer review of "Immunomodulatory Effects of Traditional Korean Gochujang in Rats Immunosuppressed with Cyclophosphamide"

_ijms, 2025, doi:10.3390/ijms26178325_

Round 1

Reviewer 1 Report

Comments and Suggestions for Authors

1.Detailed preparation process and composition analysis of supplementary samples are recommended.

2.Analyze the content of active ingredients other than capsaicin to explain the reasons for the differences in immune effects of different samples.

  1. The content of capsaicin in S4 (commercial brand) was significantly lower than that of other samples, and it was necessary to discuss whether the effect deviation was caused by additives or processes.
  2. Is weight regain associated with improved appetite in Figure 1? It is recommended to supplement the feeding intake data.
  3. The route of administration of cyclophosphamide (oral/intraperitoneal) and the basis for dosage selection should be explained.
  4. The drying method in Section 4.1 is not clear, and different drying methods may affect the retention of the active ingredient.
  5. The centrifugation conditions for serum separation in Section 4.9 are not defined and may affect cytokine stability.

Author Response

Comments 1: Detailed preparation process and composition analysis of supplementary samples are recommended.

Response 1: Section “4.2. High Performance Liquid Chromatography of GCJs”, has been revised as suggested.

Comments 2: Analyze the content of active ingredients other than capsaicin to explain the reasons for the differences in immune effects of different samples.

Response 2: It is true that gochujang contains various active components other than capsaicin. However, since red pepper powder is the primary ingredient in gochujang, capsaicin is typically the most abundant bioactive compound detected. As a fermented food, gochujang has been shown in this study to increase the levels of butyric acid, one of the key short-chain fatty acids produced by gut microbiota. The increase in butyric acid is known to contribute to improved gut health and plays a role in modulating immune cell activity, thereby helping to suppress inflammatory responses.

Whether this increase in butyric acid is directly induced by capsaicin or by other components within gochujang remains unclear. Further research is necessary to clarify which specific compounds are responsible for this effect.

Comments 3: The content of capsaicin in S4 (commercial brand) was significantly lower than that of other samples, and it was necessary to discuss whether the effect deviation was caused by additives or processes.

Response 3: The capsaicin content in sample S4 (commercial brand) was significantly lower compared to the other Gochujang. This discrepancy may be attributed to differences in manufacturing processes, including fermentation time, or the proportion of Gochu (red pepper) powder used. Additionally, commercial products often include additives such as sweeteners, thickeners, or flavor enhancers, which could dilute the concentration of bioactive compounds like capsaicin.

I agree that the observed deviation in immune-related effects may be related to these factors. However, since detailed compositional data (e.g., additive types and proportions) from the commercial product were not disclosed, it was not possible to definitively determine the cause. Further investigation, including controlled comparison of processing methods and additive content, would be required to clarify the contribution of such variables to the biological effects observed.

Comments 4: Is weight regain associated with improved appetite in Figure 1? It is recommended to supplement the feeding intake data.

Response 4: Cyclophosphamide (CP), used as an immunosuppressant, is known to induce loss of appetite, thereby reducing food intake and body weight in animal models. Accordingly, the CP-treated groups (with GCJs and HemoHim) showed a significant decrease in body weight compared with the normal group. However, food intake data were not collected in this study. In our previous research, gochujang intake was found to reduce food consumption. Therefore, the body weight gain observed in this study is unlikely to be related to food intake.

Reference: https://pubmed.ncbi.nlm.nih.gov/38515681/

Comments 5: The route of administration of cyclophosphamide (oral/intraperitoneal) and the basis for dosage selection should be explained.

Response 5: To determine the optimal conditions for inducing immunosuppression, our research team previously conducted a preliminary study using various concentrations of cyclophosphamide (CP) administered orally. Based on the outcomes of these preliminary tests, 5 mg/kg was identified as the most effective dose that consistently induced immunosuppression without causing excessive toxicity. Therefore, we established 5 mg/kg as the final dose for our study. Furthermore, a 4-week oral administration period was selected based on this initial evaluation, which demonstrated sufficient immunosuppressive effects within this timeframe. This dosing strategy is consistent with a previously published study that also reported effective immunosuppression under the same conditions.

Reference links

https://pubmed.ncbi.nlm.nih.gov/39328544/

https://www.mdpi.com/2076-3417/14/23/11013

https://onlinelibrary.wiley.com/doi/full/10.1155/2024/5010095

https://medicinalcrop.org/_PR/view/?aidx=36536&bidx=3285

Comments 6: The drying method in Section 4.1 is not clear, and different drying methods may affect the retention of the active ingredient.

Response 6: The dry weight was determined using only a portion of the sample, and it was not used in the experiment. The drying method has been revised and described in Section 4.1.

Comments 7: The centrifugation conditions for serum separation in Section 4.9 are not defined and may affect cytokine stability.

Response 7: The procedure for serum separation is already described in Section “4.6. Serum levels of cytokines and immunoglobulin G (IgG)”.

Reviewer 2 Report

Comments and Suggestions for Authors

This is an interesting and valuable study that explores the effects of 4 types of Gochujang on immunosuppression induced in Rats with cyclophosphamide. The study is important for enhancing scientific understanding of the immunomodulatory effects of fermented foods such as Gochujang.

Below are some suggestions to help improve the article:

  1. The title could be more concise. For example, 'Immunomodulatory Effects of Traditional Korean Gochujang in Rats Immunosuppressed with Cyclophosphamide (as a suggestion).
  2. In the abstract, suggest explaining briefly what Gochujang is, as was done for some of the other fermented foods mentioned.
  3. Line 23, 'increased immune related organs' suggest changing to 'increased weight of lymphoid organs' to be more specific about what actually increased.
  4. Line 24, suggest changing to 'increased numbers of white blood cells, including granulocytes and lymphocytes' so that it is clearer that granulocytes and lymphocytes are white blood cells. This improved clarity could be implemented throughout the document.
  5. Line 50 to 51, 'Accordingly, cytokine production, NK cell activity and splenocyte proliferation were investigated to assess immune enhancement [3]'. Suggest clarifying if this sentence relates to published results or the current study.
  6. In the materials and methods section, suggest explaining more clearly the treatment of the different groups. Perhaps a table could be included to make clearer the treatment of each group of rats. Also, recommend being consistent with the terminology used to describe the different groups throughout the text. 
  7. Explain what HemoHIM herbal preparation is and why it was used as a positive control. Could the results of the positive control be discussed more?
  8. Abbreviations should be written in full with the abbreviation in brackets the first time it is used, thereafter the abbreviation only can be used (please check throughout the document). 
  9. The quality of figure 1A could be improved. It is a bit difficult to differentiate the different groups.

  10. Suggest also explaining the meaning of the different letters in the statistical analysis section of the materials and methods. 
  11. INF or IFN? be consistent throughout the document. Suggest using IFN. 
  12. Use of the English language was good, but some minor improvements could be made (suggest checking the document).

Author Response

Comments 1: The title could be more concise. For example, 'Immunomodulatory Effects of Traditional Korean Gochujang in Rats Immunosuppressed with Cyclophosphamide (as a suggestion).

Response 1: Thank you for the suggestion. I have revised the sentence accordingly.

Comments 2: In the abstract, suggest explaining briefly what Gochujang is, as was done for some of the other fermented foods mentioned.

Response 2: Thank you for pointing that out. I have included a short description of Gochujang in the abstract, similar to the explanations provided for the other fermented foods.

Comments 3: Line 23, 'increased immune related organs' suggest changing to 'increased weight of lymphoid organs' to be more specific about what actually increased.

Response 3: Thank you for the suggestion. I have revised the sentence accordingly.

Comments 4: Line 24, suggest changing to 'increased numbers of white blood cells, including granulocytes and lymphocytes' so that it is clearer that granulocytes and lymphocytes are white blood cells. This improved clarity could be implemented throughout the document.

Response 4: Thank you for the suggestion. I have revised the sentence accordingly.

Comments 5: Line 50 to 51, 'Accordingly, cytokine production, NK cell activity and splenocyte proliferation were investigated to assess immune enhancement [3]'. Suggest clarifying if this sentence relates to published results or the current study.

Response 5: I have revised the sentence in lines 46–48 to clarify the indicators of immune enhancement. Based on previous studies, cytokine secretion, NK cell activity, and splenocyte proliferation are considered key markers of immune enhancement. Therefore, I have rewritten the sentence to emphasize that these three parameters are necessary to evaluate the immune-enhancing effects.

Comments 6: In the materials and methods section, suggest explaining more clearly the treatment of the different groups. Perhaps a table could be included to make clearer the treatment of each group of rats. Also, recommend being consistent with the terminology used to describe the different groups throughout the text.

Response 6: I have revised the Materials and Methods section to provide a clearer explanation of the treatment of each group, and we have also included a table to summarize the treatment protocol in “4.3 preparation of GCJs”

I reviewed the abbreviations (such as GCJ to GCJs and INF to IFN) and made the corrections

Comments 7: Explain what HemoHIM herbal preparation is and why it was used as a positive control. Could the results of the positive control be discussed more?

Response 7: In the section 4.3. Animals and oral administration, I have added the information on the supplier of HemoHIM and clarified that it was used as a positive control based on previous studies demonstrating its immune-enhancing effects.

Comments 8: Abbreviations should be written in full with the abbreviation in brackets the first time it is used, thereafter the abbreviation only can be used (please check throughout the document).

Response 8: I checked and corrected it.

Comments 9: The quality of figure 1A could be improved. It is a bit difficult to differentiate the different groups.

Response 9: Thank you for your comment regarding Figure 1A. Although it may be somewhat difficult to distinguish, we tried to maximize clarity by using different colors for each sample and by presenting the statistical annotations in the same color as the corresponding line. Specifically, the black line represents the Normal group and the red line represents the Vehicle group. The symbols "a" and "c" indicate statistical significance, as does the comparison between "a" and "b". Since all gochujang-fed groups are labeled with "b", this indicates a significant increase in body weight compared with the Vehicle group. We believe that the statistical annotations alone allow clear differentiation among the groups, and therefore we consider it appropriate not to modify the figure.

Comments 10: Suggest also explaining the meaning of the different letters in the statistical analysis section of the materials and methods.

Response 10: In SPSS statistical analysis, identical superscript letters indicate no significant difference among the corresponding groups, whereas different letters denote statistically significant differences. For instance, groups labeled “a” and “b” are significantly different from each other. In contrast, a group labeled “ab” is considered not significantly different from either group “a” or group “b,” as it shares a common superscript with both.

The above contents are summarized and inserted in the “4.11 Statistical analysis” of the manuscript.

Comments 11: INF or IFN? be consistent throughout the document. Suggest using IFN.

Response 11: I have revised it to 'IFN'.

Comments 12: Use of the English language was good, but some minor improvements could be made (suggest checking the document).

Response 12: I have carefully reviewed the manuscript and made minor improvements to the English language throughout the document.